# Solving Soft Clustering Ensemble via $k$-Sparse Discrete Wasserstein Barycenter

**Ruizhe Qin**[1]    **Mengying Li**[2]    **Hu Ding**[1]*
[1]School of Computer Science and Technology
[2]School of Data Science
University of Science and Technology of China
red46@mail.ustc.edu.cn, limengy@mail.ustc.edu.cn, huding@ustc.edu.cn

## Abstract

Clustering ensemble is one of the most important problems in ensemble learning. Though it has been extensively studied in the past decades, the existing methods often suffer from the issues like high computational complexity and the difficulty on understanding the consensus. In this paper, we study the more general soft clustering ensemble problem where each individual solution is a soft clustering. We connect it to the well-known discrete Wasserstein barycenter problem in geometry. Based on some novel geometric insights in high dimensions, we propose the sampling-based algorithms with provable quality guarantees. We also provide the systematical analysis on the consensus of our model. Finally, we conduct the experiments to evaluate our proposed algorithms.

## 1 Introduction

*Clustering* is a fundamental topic that has important applications in various areas, such as data mining, networking, and bioinformatics [34]. In the past decades, a number of different clustering objectives and algorithms have been proposed. For example, the popular *k-means* aims to partition the given data set into $k$ clusters and minimize the average squared distance from the input data to the set of cluster centers; the well known $k$-means clustering algorithms include the Lloyd's algorithm [47], $k$-means++ [5], and local search [36]. Other clustering objectives, like hierarchical clustering [49] and density-based clustering [53], are also widely used in practice.

Obviously, different clustering algorithms can obtain different results. Moreover, even for the same clustering algorithm (*e.g.,* the Lloyd's algorithm), the initialization and data preprocessing (*e.g.,* random projection [12]) steps may yield different clustering results. Therefore, a natural idea is to aggregate these different clustering results so as to achieve a more reliable result. The problem is called *clustering ensemble* (also termed *clustering aggregation* or *consensus clustering*) [27].

However, the current methods still suffer from several issues in theory and practice. Most of existing clustering ensemble methods rely on complicated optimization models, such as the correlation clustering [28], graph partition [25], semi-definite programming [54], matrix completion [63], and spectral clustering [56]; these optimization problems usually have super-linear complexities and thus cannot be efficiently solved for large-scale datasets. Though several heuristic ideas have been proposed for speeding up the computation (*e.g.,* the sampling idea proposed in [61]), they are in lack of rigorous theoretical analysis on their quality guarantees.

Another issue is about the interpretability of consensus. A large number of clustering ensemble models are based on *utility function* [57, 61, 60] or *co-association matrix* [26, 56]. From the theoretical perspective, a fundamental question is why these models can yield the final clusterings close to

---

*Corresponding author.

35th Conference on Neural Information Processing Systems (NeurIPS 2021).

the ground-truth clustering. The similar consensus question was also studied for the classification problem before [13]. However, the analysis for clustering ensemble is more challenging, because we need to take into account the matchings between different clustering solutions.

## 1.1 Our Contributions

In this paper, we focus on the more general *soft clustering ensemble* problem, where each given individual clustering is a soft clustering (also referred to as fuzzy clustering) [61]. In a soft clustering, data points can potentially belong to multiple clusters. For example, a point may be assigned to three clusters with the probabilities $10\%$, $20\%$, and $70\%$, respectively. Compared with hard clustering, soft clustering can provide more realistic and accurate clustering results in many real-world applications [9].

We adopt the geometric model that was studied in [18, 20, 19]. They showed that clustering ensemble can be naturally formulated as a "geometric prototype" problem. But their results are in lack of systematically studies on the efficiency of this model, especially from the theoretical perspective. In this paper, we illustrate that the geometric prototype actually is equivalent with an instance of *Discrete Wasserstein Barycenter (DWB)* [1] in high dimensions (the formal definition will be given in Section 3). This approach falls under the umbrella of utility function based model, where it uses discrete Wasserstein distance to measure the difference between two clusterings. Compared with other utility functions (*e.g.,* KL-divergence), it has several attractive properties. For example, the discrete Wasserstein distance is symmetric and more robust to noise [42]. More importantly, the DWB based ensemble model can be easily interpreted from the geometric perspective, and thus we can analyze its performance more conveniently. But when applying the DWB model to the soft clustering ensemble problem, we still have several key issues remaining to be solved.

**(i)** Though a number of DWB algorithms have been developed (as shown in Section 1.2), the clustering ensemble imposes two unique features to the DWB formulation. First, we require the returned DWB to be $k$-sparse, that is, it should be supported by at most $k$ points in the space (since there are at most $k$ clusters). Also, the number of different clustering solutions can be large in practical scenarios. For instance, to guarantee the consistency of the final ensemble solution to the ground-truth clustering, we may generate a large number of clustering solutions via random initializations or random projections [24]. So from the algorithmic perspective, a natural question is whether we can develop more efficient algorithm for the DWB problem with such features?

**(ii)** To the best of our knowledge, only Topchy *et al.* [58] and Jain [35] discussed the consensus of clustering ensemble in theory. However, both of their analyses rely on the assumption that the ground-truth clustering should be the optimal solution of the ensemble model, which is too strong and may not be realistic in practice. Also, the analysis of [58] only considered hard clustering. It is worth noting that the number of hard clusterings on a fixed set of items is finite, but the number of soft clusterings is infinity.

In this paper, we focus on these two issues. First, based on some novel geometric insights, we show that it is possible to achieve a *fixed-parameter algorithm* for the soft clustering ensemble problem if $k$ is a constant, where the obtained approximation factor is $1 + \epsilon$ with $\epsilon$ being any small number in $(0, 1)$; though this is more a theoretical result, we believe that it is of independent interest for such a combinatorial optimization problem in high dimensions [17]. Moreover, the proposed sampling idea inspires our following speedup for the existing DWB algorithms with provable quality guarantee, even if $k$ is large. Second, we prove that the obtained DWB should be close to the ground-truth clustering if the number of given clustering solutions is large enough. Our idea is quite different from [58, 35]; in particular, our analysis yields a detailed quantive result for the consensus.

## 1.2 Related Works

**Clustering ensemble.** Clustering ensemble was introduced by Strehl and Ghosh [55], where they formulated it as two different graph partitioning problems; one is "Instance-Based Graph Formulation (IBGF)", and the other is "Cluster-Based Graph Formulation (CBGF)". Fern and Brodley [25] later proposed a hybrid graph partitioning model for clustering ensemble. Most existing clustering ensemble models can be roughly divided into utility function based [57, 61, 60] and co-association matrix based [26, 46, 56]. The utility function based model is to find the final clustering result via maximizing the total similarities to the set of given clusterings. A co-association matrix is actually

a new representation of the data items, where each entry of the matrix indicates the similarity of a pair of data items based on the information from the given clusterings. The soft clustering ensemble problem was particularly studied in [51, 61]. For more detailed discussion about ensemble clustering, the reader is referred to the survey [27].

**Wasserstein distance.** The Wasserstein distance is defined for measuring the difference between two probability distributions; if their supports are both discrete sets, the distance is called discrete Wasserstein distance (or Earth Mover's distance) [59, 52]. Computing the discrete Wasserstein distance actually is equivalent to solving a min-cost max flow problem [2, 37]. Several more efficient discrete Wasserstein distance algorithms were proposed, such as [45, 50]. In the community of machine learning, Cuturi [15] proposed a new objective called "Sinkhorn Distance" that smoothes the transportation problem with an entropic regularization term, and it can be solved much faster than computing the exact discrete Wasserstein distance. Following Cuturi's work, several improved Sinkhorn algorithms have been proposed in recent years [23, 44, 3, 48].

**Wasserstein barycenter.** If there are $m \geq 2$ different weighted point sets, the problem of discrete Wasserstein barycenter is to compute the average pattern that minimizes the total Wasserstein distances to them [1]. The recent algorithms for computing Wasserstein barycenter include [16, 7, 29, 62, 8, 4, 43]. The computational complexity of Wasserstein barycenter was studied in [39, 11, 43]. Recently, Dognin *et al.* [21] applied the Wasserstein barycenter based ensemble method to solve several supervised learning problems.

## 2 Preliminaries

We always use $A = \{a_1, a_2, \cdots, a_n\}$ to denote the set of $n$ data items which we want to cluster.

**Definition 1 (Soft Clustering).** *Let $k \in \mathbb{Z}^+$. A $k$-soft clustering of $A$ can be represented by a set of $k$ vectors $\mathcal{C} = \{S_1, \cdots, S_k\} \subset [0,1]^n$, where each vector $S_j$ represents an individual cluster and $\sum_{j=1}^{k} S_j$ is equal to the row vector $[1, 1, \cdots, 1]$. Suppose $S_{jl}$ is the $l$-th entry of $S_j$ for $1 \leq l \leq n$. Then, for a fixed $l$, the set $\{S_{1l}, \cdots, S_{kl}\}$ indicates the degrees of membership of $a_l$ to the $k$ clusters.*

For example, suppose $n = 3$ and $k = 2$; the following is a $k$-soft clustering of $A$: $\mathcal{C} = \{S_1 = [0.1, 0.3, 0.8], S_2 = [0.9, 0.7, 0.2]\}$ (*e.g.,* the item $a_2$ belongs to the first and second clusters with probability 30% and 70%, respectively).

In Definition 1, if we restrict each $S_j$ to be binary vector, the clustering $\mathcal{C}$ will be a hard clustering. Also, if a clustering $\mathcal{C}$ has $k' < k$ clusters, we can simply add $k - k'$ dummy clusters where each dummy cluster is just a zero vector $[0, 0, \cdots, 0]$.

Following the idea of [18, 6, 20], we define the function $\Delta$ to measure the difference between two clusterings of $A$. Suppose $\mathcal{C} = \{S_1, \cdots, S_k\}$ and $\mathcal{C}' = \{S'_1, \cdots, S'_k\}$ are two different soft clusterings. We build the bipartite graph $\mathcal{G}$ from $\mathcal{C}$ and $\mathcal{C}'$ as follows: each of the two columns of $\mathcal{G}$ contains $k$ vertices corresponding to the $k$ clusters in $\mathcal{C}$ and $\mathcal{C}'$ respectively; for any pair of clusters $(S_j, S'_{j'})$ with $S_j \in \mathcal{C}$ and $S'_{j'} \in \mathcal{C}'$, there is an edge connecting their corresponding vertices in $\mathcal{G}$ with a weight equal to their squared Euclidean distance $||S_j - S'_{j'}||^2$. Then, the difference of $\mathcal{C}$ and $\mathcal{C}'$, *i.e.,* $\Delta(\mathcal{C}, \mathcal{C}')$, is the cost of the minimum weight bipartite matching of $\mathcal{G}$. The function $\Delta$ can be computed through the Hungarian algorithm [14]. Assume the minimum weight bipartite matching between $\mathcal{C}$ and $\mathcal{C}'$ yields the permutation $\pi$ of $\{1, 2, \cdots, k\}$. Namely,

$$\Delta(\mathcal{C}, \mathcal{C}') = \sum_{j=1}^{k} ||S_j - S'_{\pi(j)}||^2. \tag{1}$$

Obviously, if $\Delta(\mathcal{C}, \mathcal{C}') = 0$, they should be the same clustering solution (the $k$ clusters of $\mathcal{C}$ is just reordered by $\pi$ in $\mathcal{C}'$). To see the rationale behind (1), we can imagine the case that $\mathcal{C}$ and $\mathcal{C}'$ are both hard clusterings, *i.e.,* each $S_j \in \mathcal{C}$ (*resp.,* $S'_{j'} \in \mathcal{C}'$) is a binary vector; so each such vector can be viewed as a subset of $A$ (*e.g.,* $S_j$ represents the set $\{a_l \mid a_l \in A$ and $S_{jl} = 1\}$). It is easy to know the symmetric difference of $S_j$ and $S'_{j'}$, $|S_j \setminus S'_{j'}| + |S'_{j'} \setminus S_j|$, is equal to $||S_j - S'_{j'}||^2$. Therefore, computing the function $\Delta(\mathcal{C}, \mathcal{C}')$ in fact is to find the matching of $\mathcal{C}$ and $\mathcal{C}'$ that minimizes their total symmetric differences.

Following Definition 1, we then introduce "soft clustering ensemble" below.

**Definition 2** (**Soft Clustering Ensemble (SCE)**). *Given $m$ different soft clusterings $\mathcal{C}_1, \cdots, \mathcal{C}_m$ of $A$, the problem of soft clustering ensemble (**SCE**) is to find the final soft clustering $\tilde{\mathcal{C}}$ that minimizes the objective function*

$$\frac{1}{m} \sum_{i=1}^{m} \Delta(\tilde{\mathcal{C}}, \mathcal{C}_i). \tag{2}$$

*For any soft clustering $\tilde{\mathcal{C}}'$ and $\lambda \geq 1$, if it achieves an objective value no larger than $\lambda$ times the minimum value of (2), we say it is a "$\lambda$-approximation" for the SCE problem.*

Let $\pi_i$ be the permutation between $\tilde{\mathcal{C}}$ and $\mathcal{C}_i$ for $1 \leq i \leq m$. To minimize the objective function (2), the major challenge is to find these $m$ permutations simultaneously. Suppose each $\mathcal{C}_i = \{S_1^i, \cdots, S_k^i\}$. Once these $m$ permutations are obtained, the set $\cup_{i=1}^{m} \mathcal{C}_i$ is divided into $k$ parts:

$$\{S_{\pi_1(j)}^1, \cdots, S_{\pi_m(j)}^m\}, \quad 1 \leq j \leq k. \tag{3}$$

If we let the optimal solution $\tilde{\mathcal{C}}$ be $\{\tilde{S}_1, \cdots, \tilde{S}_k\}$, from the objective function (2) it is easy to know these $k$ soft clusters should be the means of these $k$ parts, *i.e.,*

$$\tilde{S}_j = \frac{1}{m} \sum_{i=1}^{m} S_{\pi_i(j)}^i, 1 \leq j \leq k. \tag{4}$$

This simple fact will be used in the analysis in our paper. We also have the following hardness result for SCE through the reduction from the NP-hard three-dimensional assignment problem [41].

**Theorem 1** (**The hardness**). *When $m \geq 3$, optimizing the SCE objective (2) is NP-hard.*

**The rest of the paper is organized as follows.** In Section 3, we discuss the relation between the SCE problem and discrete Wasserstein barycenter. In Section 4, we present our approximation algorithms based on random sampling. In Section 5, we analyze the consensus of the SCE problem under Definition 2. Finally, we illustrate our experimental results in Section 6. **Due to the space limit**, we leave some proofs and the detailed experimental results to the full version of this paper.

## 3 Relation With Discrete Wasserstein Barycenter

We introduce the relation between SCE and discrete Wasserstein barycenter in this section.

**Definition 3** (**Discrete Wasserstein Distance [52]**). *Let $P = \{p_1, p_2, \cdots, p_{n_P}\}$ and $Q = \{q_1, q_2, \cdots, q_{n_Q}\}$ be two sets of weighted points in $\mathbb{R}^d$ with nonnegative weights $\alpha_i$ and $\beta_j$ for each $p_i \in P$ and $q_j \in Q$, and $\sum_{i=1}^{n_P} \alpha_i = \sum_{j=1}^{n_Q} \beta_j = 1$. Their discrete Wasserstein distance is*

$$\mathcal{W}_s^s(P, Q) = \min_F \sum_{i=1}^{n_P} \sum_{j=1}^{n_Q} f_{ij} ||p_i - q_j||_s^s, \tag{5}$$

*where $|| \cdot ||_s$ indicates the $l_s$-norm and $F = \{f_{ij} \mid 1 \leq i \leq n_P, 1 \leq j \leq n_Q\}$ is a feasible flow from $P$ to $Q$, i.e., each $f_{ij} \geq 0$, $\sum_{i=1}^{n_P} f_{ij} = \beta_j$, and $\sum_{j=1}^{n_Q} f_{ij} = \alpha_i$.*

In this paper, we only focus on the $l_2$-discrete Wasserstein distance. The $l_2$-discrete Wasserstein barycenter (**DWB**$_2$) considers the following objective. Given the nonnegative weighted point sets $\{P_1, P_2, \cdots, P_m\}$ in $\mathbb{R}^d$, where each $P_i$ has the total weights equal to 1, the goal is to find a set of centroid points $\tilde{P}$, so as to minimize

$$\frac{1}{m} \sum_{i=1}^{m} \mathcal{W}_2^2(P_i, \tilde{P}). \tag{6}$$

Further, if we require $\tilde{P}$ to have at most $k$ points with some $k \in \mathbb{Z}^+$, the problem is called "$k$-**sparse DWB**$_2$" [11]. It is easy to observe that the $k$-sparse DWB$_2$ problem is very similar to SCE as described in Definition 2. Intuitively, the obtained optimal ensemble clustering $\tilde{\mathcal{C}}$ can be viewed as the $k$-sparse DWB$_2$, if each point of $\mathcal{C}_i$ is assigned the weight $1/k$ for $1 \leq i \leq m$.

**Theorem 2.** *Given a set of soft clusterings $\{\mathcal{C}_1, \cdots, \mathcal{C}_m\}$, the optimal solution of the SCE problem (2) is exactly their optimal $k$-sparse DWB$_2$. Moreover, for any $\lambda \geq 1$, a $\lambda$-approximation of the $k$-sparse DWB$_2$ can yield a $\lambda$-approximate solution of (2).*

To prove the above theorem, we need to consider two issues. First, the discrete Wasserstein distance of Definition 3 may result in a many-to-many matching between $\tilde{C}$ and $\mathcal{C}_i$, but the difference function $\Delta(\tilde{\mathcal{C}}, \mathcal{C}_i)$ requires a one-to-one matching. Since the discrete Wasserstein distance is actually an instance of the min-cost max flow problem, we can convert the obtained many-to-many matching to a one-to-one matching without increasing the complexity. Second, we should further prove that the obtained (optimal or approximate) $k$-sparse DWB$_2$ is a feasible soft clustering (*i.e.,* it should satisfy the conditions described in Definition 1).

## 4 Approximation Algorithms

Due to Theorem 1, we only focus on the approximation algorithms for SCE. It is also worth noting that when $m = 2$, the optimal $k$-sparse DWB$_2$ (and the SCE solution) can be easily obtained by computing the matching between $\mathcal{C}_1$ and $\mathcal{C}_2$; the optimal barycenter should just be the midpoints of the $k$ matched pairs. In this section, we propose the approximation algorithms for optimizing the SCE objective (2) with $m \geq 3$.

First, we present the following lemma which is the key to our algorithms. Given a point set $Q \subset \mathbb{R}^d$, we use $\mu(Q)$ and $\mathtt{Var}(Q)$ to denote the mean and variance respectively, *i.e.,* $\mu(Q) = \frac{1}{|Q|} \sum_{q \in Q} q$ and $\mathtt{Var}(Q) = \frac{1}{|Q|} \sum_{q \in Q} ||q - \mu(Q)||^2$.

**Lemma 1.** *[33] Let $\delta \in (0, 1)$. Given a point set $Q$, we suppose that $Q'$ is a set of $t$ points sampled from $Q$ uniformly at random. Then with probability $1 - \delta$, $||\mu(Q) - \mu(Q')||^2 \leq \frac{1}{\delta t} \mathtt{Var}(Q)$.*

**Remark 1.** *Lemma 1 reveals that we can estimate $\mu(Q)$ by just simple random sampling. For example, if we require the error no larger than $\epsilon \mathtt{Var}(Q)$ with some small $\epsilon > 0$, we only need to sample $\frac{1}{\delta \epsilon}$ points from $Q$ and the success probability is $1 - \delta$. Obviously, the smaller $\epsilon$ and $\delta$, the larger the required sample size. Moreover, a highlight of Lemma 1 is that the sample size is independent of the dimensionality.*

We also need the following lemma from [40]. Lemma 2 indicates that if a point $p$ is close to $\mu(Q)$, the total squared distances to the points of $Q$ should be close to $\mathtt{Var}(Q)$ as well.

**Lemma 2.** *Let $Q$ be a set of points in $\mathbb{R}^d$. For any point $p \in \mathbb{R}^d$, $\frac{1}{|Q|} \sum_{q \in Q} ||q - p||^2 = \mathtt{Var}(Q) + ||\mu(Q) - p||^2$.*

In Section 4.1, we propose a fixed-parameter algorithm that returns a $(1 + \epsilon)$-approximate solution, if $k$ is assumed to be small. Though it is more a theoretical result, the proposed sampling idea inspires our following improvement in Section 4.2 on the existing alternating minimization algorithms for the case that $k$ is not a constant.

### 4.1 A Fixed-parameter Algorithm

When $k$ is small, we can achieve a $(1 + \epsilon)$-approximation for the SCE problem. We briefly illustrate our idea below.

For ease of presentation, we "temporarily" assume the permutations $\pi_i$ between $\tilde{\mathcal{C}}$ and $\mathcal{C}_i$, $1 \leq i \leq m$, which yield the optimal solution of (2), are given at this moment. For each $\mathcal{C}_i$, we concatenate its $k$ vectors $\{S_1^i, \cdots, S_k^i\}$ to be a long vector

$$v_i = (S_{\pi_i(1)}^i \cdots S_{\pi_i(k)}^i) \qquad (7)$$

in $\mathbb{R}^{kn}$. See Figure 1 for an illustration. Meanwhile, the ensemble clustering $\tilde{\mathcal{C}} = \{\tilde{S}_1, \cdots, \tilde{S}_k\}$ can be also represented as a vector

Figure 1: An illustration for the vector $v_i$.

$$\tilde{v} = (\tilde{S}_1 \cdots \tilde{S}_k) \qquad (8)$$

**Algorithm 1** $(1 + \epsilon)$-Approximate SCE Algorithm

---

**Input:** $m$ $k$-soft clusterings $\{\mathcal{C}_1, \mathcal{C}_2, \cdots, \mathcal{C}_m\}$ on a set $A$ of $n$ data items, and two parameters $0 < \delta, \epsilon \le 1$.

  1. Select $t = \frac{1}{\delta\epsilon}$ clusterings uniformly at random, and denote them by $\mathcal{C}_{i_1}, \mathcal{C}_{i_2}, \cdots, \mathcal{C}_{i_t}$. For $1 \le l \le t$, $\mathcal{C}_{i_l}$ contains $k$ vectors (*i.e.*, soft clusters) $\{S_1^{i_l}, \cdots, S_k^{i_l}\}$ in $[0, 1]^n$.
  2. Initiate a candidate set $\mathbb{G} = \varnothing$.
  3. For each $\mathcal{C}_{i_l}$, $1 \le l \le t$, enumerate all the $k!$ possible permutations for $\pi_{i_l}$ (so there are $(k!)^t$ cases in total). For each case:

       (a) Let $\mathcal{C} = \{S_1, S_2, \cdots, S_k\}$, where each

$$S_j = \frac{1}{t} \sum_{l=1}^{t} S_{\pi_{i_l}(j)}^{i_l}, j = 1, 2, \cdots, k.$$

       (b) Update $\mathbb{G} = \mathbb{G} \cup \{\mathcal{C}\}$.

  4. For each candidate $\mathcal{C} \in \mathbb{G}$, compute the objective value $\frac{1}{m} \sum\limits_{i=1}^{m} \Delta(\mathcal{C}, \mathcal{C}_i)$. Let $\bar{\mathcal{C}}$ be the one with the smallest objective value among $\mathbb{G}$.

**Output:** $\bar{\mathcal{C}}$ as the final solution.

---

in $\mathbb{R}^{kn}$. We show that $\tilde{v}$ in fact is the mean of $\{v_1, \cdots, v_m\}$. Then we can apply Lemma 1 to estimate the position of $\tilde{v}$. Finally, since $k$ is assumed to be a constant, we can enumerate all the possible permutations of the sampled clusterings and choose the best one as the final solution. Though Theorem 3 is more a theoretical result, we believe that it is of independent interest as a fixed-parameter solution for such a combinatorial optimization problem in high dimensions.

**Theorem 3.** *(i) With probability $1 - \delta$, Algorithm 1 yields a $(1 + \epsilon)$-approximate solution of the SCE problem. (ii) The runtime is $O\big(\exp(\frac{1}{\delta\epsilon}k \log k) \cdot k^2 \cdot mn\big)$*

*Proof.* First, we need to show that each candidate $\mathcal{C} \in \mathbb{G}$ is a feasible soft clustering, that is, its $k$ vectors $\{S_1, \cdots, S_k\}$ generated in Step 3(a) should satisfy the conditions of Definition 1. Since each $S_j = \frac{1}{t} \sum_{l=1}^{t} S_{\pi_{i_l}(j)}^{i_l}$, it must belong to $[0, 1]^n$. Also,

$$\sum_{j=1}^{k} S_j = \sum_{j=1}^{k} \frac{1}{t} \sum_{l=1}^{t} S_{\pi_{i_l}(j)}^{i_l} = \frac{1}{t} \sum_{l=1}^{t} \sum_{j=1}^{k} S_{\pi_{i_l}(j)}^{i_l} = [1, 1 \cdots, 1], \tag{9}$$

where the final equality comes from the fact that each sampled $\mathcal{C}_{i_l}$ is a feasible soft clustering. Consequently, each candidate $\mathcal{C}$ from $\mathbb{G}$ is also a feasible soft clustering.

Now, we consider the induced objective value of the best candidate selected from $\mathbb{G}$. As discussed before (see Figure 1), each soft clustering can be converted to a long vector in $\mathbb{R}^{kn}$. Denote by $v_i$ the vector of $\mathcal{C}_i$, for $1 \le i \le m$; similarly, denote by $\tilde{v}$ the vector of the optimal solution $\tilde{\mathcal{C}}$. We use $V$ to denote the set $\{v_1, \cdots, v_m\}$. Obviously, $\tilde{v} = \mu(V)$ (from the fact (4)). Since $\{\mathcal{C}_{i_1}, \mathcal{C}_{i_2}, \cdots, \mathcal{C}_{i_t}\}$ are the randomly selected $t = \frac{1}{\delta\epsilon}$ clusterings from the input, together with Lemma 1, we have

$$||\tilde{v} - \tilde{v}'||^2 \le \epsilon \texttt{Var}(V) \tag{10}$$

with probability $1 - \delta$, where $\tilde{v}' = \frac{1}{t} \sum_{l=1}^{t} v_{i_l}$. Since $\frac{1}{m} \sum_{i=1}^{m} ||v_i - \tilde{v}'||^2 = \texttt{Var}(V) + ||\tilde{v} - \tilde{v}'||^2$ (by Lemma 2), the inequality (10) implies

$$\frac{1}{m} \sum_{i=1}^{m} ||v_i - \tilde{v}'||^2 \le (1 + \epsilon)\texttt{Var}(V). \tag{11}$$

Hence, $\tilde{v}'$ is a $(1 + \epsilon)$-approximate solution of the objective (2). Because we do not know those permutations $\pi_{i_l}$, $1 \le l \le t$, we cannot directly obtain $\tilde{v}'$. Through enumerating all the $(k!)^t$ cases, we can claim that there must exist one candidate in $\mathbb{G}$ that yields a $(1 + \epsilon)$-approximation.

The time complexity contains two parts, *i.e.*, constructing $|\mathbb{G}|$ and selecting the best candidate from $\mathbb{G}$. Since $|\mathbb{G}| = (k!)^t = O(\exp(\frac{1}{\delta\epsilon}k \log k))$, the first part takes $O(\exp(\frac{1}{\delta\epsilon}k \log k) \cdot t \cdot kn)$ time.

For the second part, we use the Hungarian algorithm to compute the one-to-one matching from $\mathcal{C}_i$, $1 \leq i \leq m$, to each candidate of $\mathbb{G}$. Note that the complexity of the Hungarian algorithm is $O(k^3 + k^2 \cdot n) = O(k^2 n)$ ($n \gg k$), where the term $k^2 n$ is due to the time for building the $k \times k$ bipartite graph in $\mathbb{R}^n$. Therefore the second part takes $O(\exp(\frac{1}{\delta\epsilon} k \log k) \cdot k^2 \cdot mn)$ time. Overall, the second part dominates the whole complexity, and the runtime of Algorithm 1 is $O(\exp(\frac{1}{\delta\epsilon} k \log k) \cdot k^2 \cdot mn)$. □

## 4.2 When $k$ Is Not a Constant

Now we consider the case that $k$ is not a constant. Due to the discussion in Section 3, we can directly apply any existing $k$-sparse DWB$_2$ algorithm to compute the final ensemble clustering. An interesting observation is that a couple of widely used DWB$_2$ algorithms follow the alternating minimization framework [16, 7, 62] (several previous ensemble clustering algorithms also used this alternating minimization idea [31, 20]): in each iteration, the algorithm alternatively performs the following two steps:

1. update the matchings (*i.e.,* the Wasserstein flows) from those $\mathcal{C}_i$s to the temporary barycenter;
2. update the $k$ points of the temporary barycenter based on the new matchings.

Different algorithms may adopt different strategies for implementing these two steps. Eventually, the result will converge to a local optimum (note it is NP-hard to achieve a global optimum according to Theorem 1). A bottleneck of this framework, especially when $m$ is large, is the computation for the Wasserstein distances over all the $m$ clusterings $\mathcal{C}_1, \cdots, \mathcal{C}_m$. In fact, similar to Algorithm 1, we can still apply Lemma 1 to reduce the time complexity for this bottleneck. Let $\tilde{\mathcal{C}}_\perp$ be the temporary barycenter at the beginning of the current iteration. Suppose $\tilde{\mathcal{C}}_\top$ is the updated barycenter if we compute all the $m$ Wasserstein distances. Let $0 \leq \epsilon, \delta \leq 1$. If we randomly select $\frac{1}{\epsilon\delta}$ clusterings and only compute these $\frac{1}{\epsilon\delta}$ Wasserstein distances to update $\tilde{\mathcal{C}}_\perp$ to be $\tilde{\mathcal{C}}'_\top$, we have the following result via the same idea of Theorem 3.

**Lemma 3.** *With probability $1 - \delta$, $\frac{1}{m} \sum_{i=1}^{m} \Delta(\tilde{\mathcal{C}}'_\top, \mathcal{C}_i) \leq (1 + \epsilon) \cdot \frac{1}{m} \sum_{i=1}^{m} \Delta(\tilde{\mathcal{C}}_\top, \mathcal{C}_i)$.*

Lemma 3 indicates that we can avoid computing all the $m$ Wasserstein distances and still achieve a barycenter close to $\tilde{\mathcal{C}}_\top$ via random sampling. **Note** that a key difference with Algorithm 1 is that we do not need to enumerate all the permutations since the $\frac{1}{\epsilon\delta}$ matchings can be determined by $\tilde{\mathcal{C}}_\perp$.

## 5 Analysis on the Consensus

Denote by $\mathcal{C}_{\text{gt}}$ the ground-truth clustering over the $n$ data items of $A$. In this section, we analyze the consensus of the objective (2). Specifically, when the number $m$ increases, we are wondering whether the optimal solution $\tilde{\mathcal{C}}$ will converge to the ground-truth clustering $\mathcal{C}_{\text{gt}}$. Let $\Omega$ be the set of all the possible soft clusterings over $A$, following a probability measure $\rho$. Namely, for any soft clustering $\mathcal{C} \in \Omega$, the probability of obtaining it is $\rho(\mathcal{C})$, and $\int_{\mathcal{C} \in \Omega} \rho(\mathcal{C}) d\mathcal{C} = 1$.

First, we propose the following assumption that there is an upper bound for the difference between $\mathcal{C}_{\text{gt}}$ and any $\mathcal{C} \in \Omega$.

**Assumption 1.** *There exists a value $L > 0$, such that $\Delta(\mathcal{C}, \mathcal{C}_{\text{gt}}) \leq L$ for any $\mathcal{C} \in \Omega$.*

Actually, this assumption is easy to understand in the context of clustering ensemble. Since each clustering solution $\mathcal{C}$ is obtained by some reasonable clustering algorithm, it makes sense to assume that these solutions are not arbitrarily far from $\mathcal{C}_{\text{gt}}$. Assumption 1 directly implies the following Lemma which can be proved by the fact $\Delta(\mathcal{C}, \mathcal{C}') \leq 2\Delta(\mathcal{C}, \mathcal{C}_{\text{gt}}) + 2\Delta(\mathcal{C}', \mathcal{C}_{\text{gt}})$.

**Lemma 4.** *For any $\mathcal{C}$ and $\mathcal{C}' \in \Omega$, $\Delta(\mathcal{C}, \mathcal{C}') \leq 4L$.*

As mentioned in Section 1.1, to the best of our knowledge, only Topchy *et al.* [58] and Jain [35] discussed the consensus of clustering ensemble in theory. However, both of their analyses need the assumption that the ground-truth clustering should be exactly the one achieving the smallest expected objective value over all the possible clustering solutions:

$$\mathcal{C}_{\text{gt}} = \arg\min_{\hat{\mathcal{C}} \in \Omega} \int_\Omega \rho(\mathcal{C}) \Delta(\hat{\mathcal{C}}, \mathcal{C}) d\mathcal{C}, \tag{12}$$

which may be too strong in reality. Also, they did not provide the quantitive analysis, *e.g.,* the numerical relation between the convergence and the value $m$. Here, we relax the assumption (12). In particular, we allow $\mathcal{C}_{\text{gt}} \neq \arg\min_{\hat{\mathcal{C}} \in \Omega} \int_{\Omega} \rho(\mathcal{C}) \Delta(\hat{\mathcal{C}}, \mathcal{C}) \mathrm{d}\mathcal{C}$. We instead assume any clustering solution that achieves sufficiently low expected objective value, should be close to $\mathcal{C}_{\text{gt}}$. Let $\mathtt{Opt} := \min_{\hat{\mathcal{C}} \in \Omega} \int_{\Omega} \rho(\mathcal{C}) \Delta(\hat{\mathcal{C}}, \mathcal{C}) \mathrm{d}\mathcal{C}$, and $\mathcal{C}_{\text{opt}}$ be the clustering solution that achieves $\mathtt{Opt}$. Note that $\mathcal{C}_{\text{opt}}$ and $\mathcal{C}_{\text{gt}}$ are not necessary to be the same one, and our ultimate goal is to find a solution close to $\mathcal{C}_{\text{gt}}$.

**Assumption 2.** *There exist two numbers $c, \xi \geq 0$, such that for any $\hat{\mathcal{C}} \in \Omega$, if its expected objective value $\int_{\Omega} \rho(\mathcal{C}) \Delta(\hat{\mathcal{C}}, \mathcal{C}) \mathrm{d}\mathcal{C} \leq (1 + c)\mathtt{Opt}$, we have $\Delta(\hat{\mathcal{C}}, \mathcal{C}_{\text{gt}}) \leq \xi$.*

Obviously, when $c = \xi = 0$, Assumption 2 will be as same as the assumption (12) from [58]. So our assumption is more relaxed. We also need to point out that the objective (2) (and the $k$-sparse Wasserstein barycenter) may have multiple isolated global optimums. But under Assumption 1, we restrict $\Omega$ to a local region and thus it is reasonable to assume Assumption 2 to be true.

To begin analyzing the consensus, we fix a clustering $\hat{\mathcal{C}} \in \Omega$ first. We let $x_i = \Delta(\hat{\mathcal{C}}, \mathcal{C}_i)$ for $i = 1, 2, \cdots, m$, and view each $x_i$ as an independent random variable. Then, we denote their mean $\frac{1}{m} \sum_{i=1}^{m} x_i$ as $\bar{x}$. Obviously, $\mathrm{E}[\bar{x}] = \int_{\Omega} \rho(\mathcal{C}) \Delta(\hat{\mathcal{C}}, \mathcal{C}) \mathrm{d}\mathcal{C}$. We then study the difference between $\bar{x}$ and $\mathrm{E}[\bar{x}]$. From Lemma 4, we know $x_i \in [0, 4L]$ for $1 \leq i \leq m$. Through the Hoeffding's inequality [30], for any $\eta > 0$ we have

$$\mathtt{Prob}\big[|\bar{x} - \mathrm{E}[\bar{x}]| > \eta \mathrm{E}[\bar{x}]\big] < 2\exp(-\frac{m\eta^2 (\mathrm{E}[\bar{x}])^2}{2L^2}).$$

Consequently, we have the following Lemma.

**Lemma 5.** *Fix $\hat{\mathcal{C}} \in \Omega$. For any $\eta, \delta \in (0, 1)$, if $m \geq \frac{8L^2}{\eta^2 (\mathrm{E}[\bar{x}])^2} \log \frac{2}{\delta}$, with probability $1 - \delta$, $\frac{1}{m} \sum_{i=1}^{m} \Delta(\hat{\mathcal{C}}, \mathcal{C}_i) \in (1 \pm \eta) \int_{\Omega} \rho(\mathcal{C}) \Delta(\hat{\mathcal{C}}, \mathcal{C}) \mathrm{d}\mathcal{C}$.*

But Lemma 5 is only for a fixed $\hat{\mathcal{C}}$. We need to extend the Lemma to any $\hat{\mathcal{C}} \in \Omega$. To realize this goal, we discretize the set $\Omega$ first.

**Discretization.** We use $\mathbb{B}(p, r)$ to denote the ball centered at a point $p$ with radius $r \geq 0$ in $\mathbb{R}^n$. We suppose the $k$ soft clusters (*i.e.,* the $k$ vectors in $\mathbb{R}^n$) of $\mathcal{C}_{\text{gt}}$ are $S_{\text{gt},1}, \cdots, S_{\text{gt},k}$. From Assumption 1, we know for each $\hat{\mathcal{C}} \in \Omega$, its $k$ vectors should be covered by the region $\mathcal{R} = \cup_{j=1}^{k} \mathbb{B}(S_{\text{gt},j}, \sqrt{L})$. Imagine that we draw a uniform grid inside $\mathcal{R}$ with the grid side length being equal to $\frac{\vartheta}{\sqrt{n}}$ (the value of $\vartheta$ will be determine later). Thus, for any two points inside the same cell of the grid, their distance is no larger than $\sqrt{n \cdot (\vartheta/\sqrt{n})^2} = \vartheta$. Moreover, by using the formula for ball volume in $\mathbb{R}^n$, we know the size of $\Gamma_j$, which denotes the set of the grid points inside $\mathbb{B}(S_{gt,j}, \sqrt{L})$, is $O\big(\big(\frac{\sqrt{\pi e L}}{\vartheta}\big)^n\big)$.

So the set $\Omega_{\text{grid}} := \Gamma_1 \times \Gamma_2 \times \cdots \times \Gamma_k$ contains $N = O\big(\big(\frac{\sqrt{\pi e L}}{\vartheta}\big)^{kn}\big)$ different $k$-tuple points (*i.e.,* soft clusterings). If we replace $\delta$ by $\delta/N$ in Lemma 5 and take the union bound over all the soft clusterings of $\Omega_{\text{grid}}$, we can obtain the following result.

**Lemma 6.** *For any $\eta, \delta \in (0, 1)$, if $m \geq O\big(\frac{knL^2}{\eta^2 (\mathtt{Opt})^2} \log \frac{L}{\vartheta \delta}\big)$, with probability $1 - \delta$, $\frac{1}{m} \sum_{i=1}^{m} \Delta(\hat{\mathcal{C}}, \mathcal{C}_i) \in (1 \pm \eta) \int_{\Omega} \rho(\mathcal{C}) \Delta(\hat{\mathcal{C}}, \mathcal{C}) \mathrm{d}\mathcal{C}$ for any $\hat{\mathcal{C}} \in \Omega_{\text{grid}}$.*

**Remark 2.** *We replace $\mathrm{E}[\bar{x}]$ by $\mathtt{Opt}$ in the lower bound of $m$ in Lemma 6, since $\mathrm{E}[\bar{x}]$ is always no smaller than $\mathtt{Opt}$.*

Now, we consider the clusterings in $\Omega \setminus \Omega_{\text{grid}}$. For each point $p \in \mathcal{R}$, we use $\mathcal{N}(p)$ to denote its nearest grid point. Similarly, for any $\hat{\mathcal{C}} = \{\hat{S}_1, \cdots, \hat{S}_k\} \in \Omega \setminus \Omega_{\text{grid}}$, we denote its "nearest clustering" as $\mathcal{N}(\hat{\mathcal{C}})$, which contains the $k$ vectors $\{\mathcal{N}(\hat{S}_1), \cdots, \mathcal{N}(\hat{S}_k)\}$. We should prove that $\big|\Delta(\mathcal{N}(\hat{\mathcal{C}}), \mathcal{C}) - \Delta(\hat{\mathcal{C}}, \mathcal{C})\big|$ is small for any $\mathcal{C} \in \Omega$, as long as $\vartheta$ is sufficiently small. Consequently, we can extend the result of Lemma 6 from $\Omega_{\text{grid}}$ to all the clusterings in $\Omega$.

**Lemma 7.** *For any $\eta, \delta \in (0, 1)$, if $m \geq O\big(\frac{knL^2}{\eta^2 (\mathtt{Opt})^2} \log \frac{kL}{\eta \delta \mathtt{Opt}}\big)$, with probability $1 - \delta$, $\frac{1}{m} \sum_{i=1}^{m} \Delta(\hat{\mathcal{C}}, \mathcal{C}_i) \in (1 \pm 7\eta) \int_{\Omega} \rho(\mathcal{C}) \Delta(\hat{\mathcal{C}}, \mathcal{C}) \mathrm{d}\mathcal{C}$ for any $\hat{\mathcal{C}} \in \Omega$.*

The value of $\vartheta$ is set to be $\frac{\eta \cdot \mathtt{Opt}}{\sqrt{8kL}}$ in the proof of Lemma 7 (the detailed proof is placed to our full paper). Finally, we achieve the consensus theorem under Assumption 1 and 2.

**Theorem 4 (Consensus).** *Let $\delta \in (0,1)$ and $\eta < \frac{c}{7(2+c)}$. Suppose $\{\mathcal{C}_1, \cdots, \mathcal{C}_m\}$ are drawn* i.i.d *from $\Omega$ with the probability $\rho(\cdot)$. If $m \geq O\left(\frac{knL^2}{\eta^2(\mathtt{Opt})^2} \log \frac{kL}{\eta \delta \mathtt{Opt}}\right)$, with probability $1 - \delta$, $\Delta(\tilde{\mathcal{C}}, \mathcal{C}_{\mathtt{gt}})$ is no larger than $\xi$, where $\tilde{\mathcal{C}}$ is the optimal solution for the objective (2).*

*Proof.* Suppose $\Delta(\tilde{\mathcal{C}}, \mathcal{C}_{\mathtt{gt}}) > \xi$. From Assumption 2, we know $\int_\Omega \rho(\mathcal{C}) \Delta(\tilde{\mathcal{C}}, \mathcal{C}) \mathrm{d}\mathcal{C} > (1 + c)\mathtt{Opt}$, which implies

$$\frac{1}{m} \sum_{i=1}^m \Delta(\tilde{\mathcal{C}}, \mathcal{C}_i) \geq (1 - 7\eta)(1 + c)\mathtt{Opt} \tag{13}$$

via Lemma 7. Moreover, by using Lemma 7 again, we have

$$\frac{1}{m} \sum_{i=1}^m \Delta(\mathcal{C}_{\mathtt{opt}}, \mathcal{C}_i) \leq (1 + 7\eta)\mathtt{Opt}. \tag{14}$$

Since we let $\eta < \frac{c}{7(2+c)}$, (13) and (14) together imply $\frac{1}{m} \sum_{i=1}^m \Delta(\mathcal{C}_{\mathtt{opt}}, \mathcal{C}_i) < \frac{1}{m} \sum_{i=1}^m \Delta(\tilde{\mathcal{C}}, \mathcal{C}_i)$, which is contradict with the fact that $\tilde{\mathcal{C}}$ is the optimal solution for the objective (2). Hence the inequality $\Delta(\tilde{\mathcal{C}}, \mathcal{C}_{\mathtt{gt}}) \leq \xi$ is true. $\qquad\square$

## 6 Experimental Results

We evaluate the practical performance of our proposed algorithm in this section. All the experimental results were obtained on a server equipped with 2.8GHz Intel CPU, 8GB main memory, and Matlab 2019a. We consider three real datasets: USPS has 11000 data items in $\mathbb{R}^{256}$ with $k = 10$ [32]; IRIS [22] has 150 data items in $\mathbb{R}^4$ with $k = 3$; CIFAR-10 [38] has 10000 data items in $\mathbb{R}^{3072}$ with $k = 10$. Similar with [24, 13], we apply random projections to generate the clustering solutions (in each random subspace, we use $k$-means to cluster the data). We consider two representative baselines: the bipartite graph partition method BGP [25]; the top-down method FURTHEST [28]. Our sampling idea of Section 4.2 is incorporated into the alternating minimization Wasserstein barycenter algorithm [62], which is denoted as AM-$r$ with $r$ representing the sample rate (*e.g.,* AM-1 means we directly run the algorithm on the original data without sampling).

We set $m = 1000$ (*i.e.,* the number of generated clustering solutions for ensemble) and show the results in Figure 2 (a)-(i). We can see our Wasserstein barycenter based algorithm significantly outperforms other baselines in terms of the objective value (2), the Wasserstein distance to ground truth, and the runtime on the datasets USPS and CIFAR-10. Only for the smallest dataset IRIS, the baselines are faster (actually all the runtimes are very close for this small dataset). Also, we study the convergence of our obtained result, *i.e.,* its Wasserstein distance to the ground-truth clustering as $m$ increases, in Figure 2 (j)-(l). We can see in general the convergence performs better when the sample rate is larger. Due to the space limit, we leave the detailed experimental results to our full paper.

## 7 Conclusion and Future Work

In this paper, we connect the soft clustering ensemble problem to $k$-sparse discrete Wasserstein barycenter. There are several interesting problems deserved to study in future. For example, the robustness of the Wasserstein barycenter based clustering ensemble is in lack of discussions so far. In particular, we can consider its robustness under adversarial attacks, *e.g.,* the poisoning and evasion attacks [10]. Also, we believe it is important to study some other relevant issues, *e.g.,* the privacy-preserving problem and the fairness problem, for clustering ensemble.

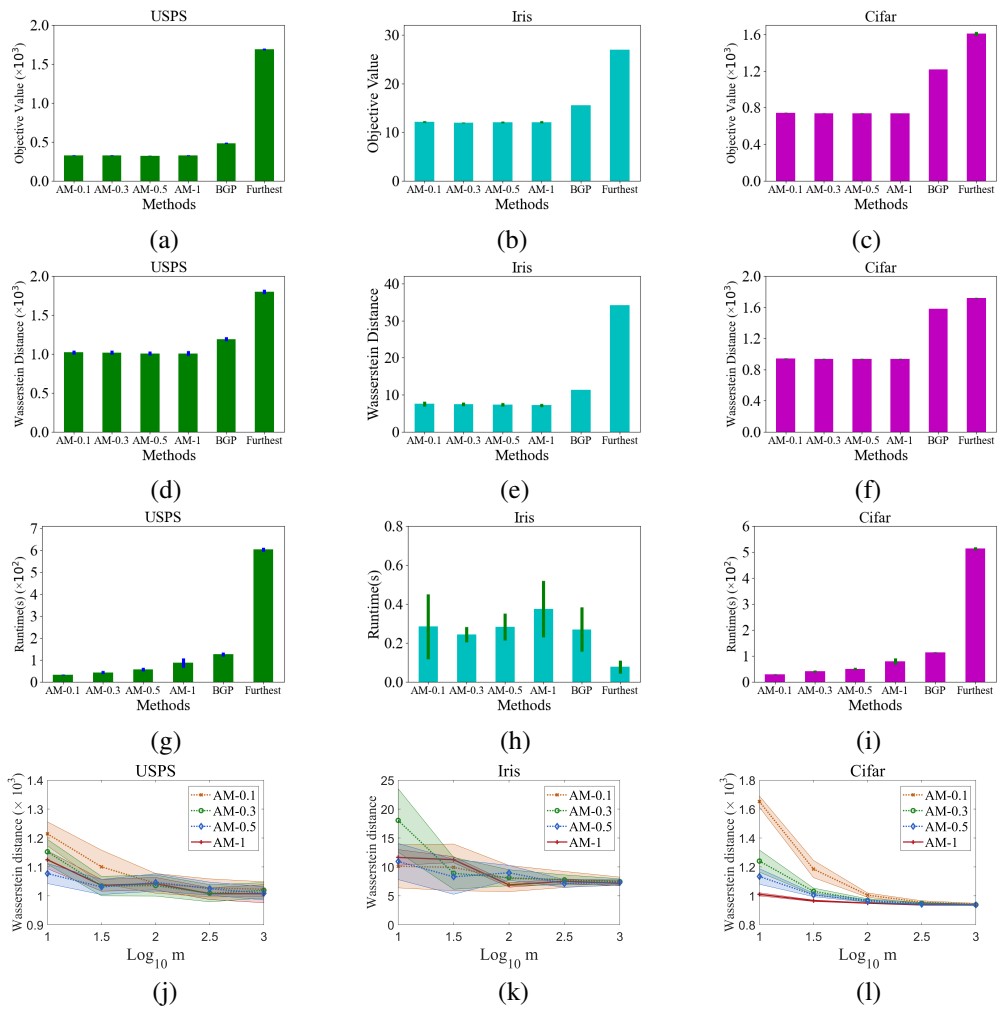

Figure 2: The objective values (the first line), the Wasserstein distance to ground truth (the second line), the runtimes (the third line), and the convergence (the third line) on the datasets. All the results are averaged across 30 trials.

## Acknowledgment

We would like to thank the anonymous reviewers for their helpful suggestions and comments.

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
