# OpenReview forum: "Solving Soft Clustering Ensemble via $k$-Sparse Discrete Wasserstein Barycenter"
_NeurIPS.cc/2021/Conference — NeurIPS 2021 Poster_

### Official Review · Reviewer_V1Ay · 2021-07-16

**Rating:** 5
**Confidence:** 3

**Summary:**

Authors connected the soft clustering ensemble and discrete Wasserstein barycenters. They proposed a (1+eps) approximation algorithm and adopted a uniform sampling scheme to reduce the time complexity of computing Wasserstein barycenters. Convergence is proved and verified via experiments.

**Limitations And Societal Impact:**

Yes.

**Main Review:**

Strengths:
+ The connection of Wasserstein barycenters and soft clustering ensemble

+ Solid the mathematical derivation seems; claims in the paper seem to be supported by either proofs or reference.

Weaknesses:
- Theorem 2 seems a straightforward result of connecting the objectives of DWB and SCE.

- Algorithm 1 is not practical ($k!$ in step 3) and not used in the experiments. Please correct me if I am wrong but I found in the code that alternative optimization for DWB was actually adopted instead.

- The results don't show much improvement for a smaller sampling rate. And in Figure 2, (a-f) I wonder if x-ticks are wrong because they suggest a large rate leads to shorter time but the largest rate AM-1 took the longest time.

**Time Spent Reviewing:**

3

---

> ### Author Response · Authors · 2021-08-10
> **Response to Reviewer V1Ay**
>
> We thank the reviewer for the thoughtful comments and suggestions. Our responses and clarifications are summarized below.
>
> **----“Theorem 2 seems a straightforward result……”**
>
> The key part for proving Theorem 2 is to show that any obtained  “$\lambda$-approximate Wasserstein barycenter” can yield a qualified soft clustering (satisfying the conditions in Definition 1). Actually, this is not that straightforward, because the obtained (approximate) Wasserstein barycenter may not satisfy the conditions in Definition 1. Our proof (in the supplement) shows that  “after some modification” on the obtained $\lambda$-approximate Wasserstein barycenter, it can yield a qualified soft clustering with the same approximation factor  “$\lambda$”.
>
> **---“…….I found in the code that alternative optimization for DWB was actually adopted instead.”**
>
> The k! permutations are computed only in our first algorithm (i.e., Algorithm 1 in section 4.1). We admit this algorithm is not that practical (that’s why it is called a “fixed-parameter algorithm”), but it inspires our speedup for the existing DWB algorithms with provable quality guarantee (please see lines 72-75). In our experiments, we apply our sampling idea of Section 4.2 to the alternating minimization WB algorithm (line 342-343). We do not need to enumerate all the k! permutations (in each iteration, the temporary barycenter from the last iteration can determine the matchings).
>
> **----“……..And in Figure 2, (a-f) I wonder if x-ticks are wrong because they suggest a large rate leads to shorter time but the largest rate AM-1 took the longest time. ”**
>
> We apologize for this confusion. We should re-order the columns (move backward AM-1 to be after AM-0.5). The “r” in “AM-r” indicates the sample rate, and AM-1 means we directly take the whole data (line 343-345). There is an increasing trend of the runtime from AM-0.1 to AM-1, but the trend looks not that significant in the figure because the runtimes are all much lower than the runtime of the baseline “Furthest”. Thanks for   this question,  and we will use log (runtime) instead in y-axis.

---

> > ### Comment · Reviewer_V1Ay · 2021-08-19
> > **Thanks for the response**
> >
> > Thank you for your response.
> >
> > I think the paper is in general very clear. My main concerns about Theorem 2 and Algorithm 1 still remain. I appreciate your explanation but I still think the contribution of the paper is limited because Theorem 2 is pretty straightforward and Algorithm 1 is not practical. The sampling idea is also not significant.
> >
> > I am keeping my original score (below threshold) at this point.
> >
> > Thank you.
> >
> > V1Ay.

---

### Official Review · Reviewer_Jegc · 2021-07-17

**Rating:** 7
**Confidence:** 3

**Summary:**

This paper studies the ensemble clustering problem with soft clustering indicators. They reveal an interesting equivalence between the soft clustering ensemble problem and the $k$-sparse discrete Wasserstein barycenter. The authors proved the hardness result of SCE and reported approximation algorithms for two different regimes (fixed $k$ or not). On the modeling side, they provided consensus results for the SCE problem under a weaker assumption on the ground-truth clustering set than prior results. They also conducted numerical experiments to support their results.

**Limitations And Societal Impact:**

The algorithm in Section 4 might run slowly in practice due to the number of permutations in the sub-procedure. In Theorem 3, the exponential dependence on $\frac{1}{\delta\epsilon}$ seems not desirable.

* Eq.(3): $W_s(P,Q)$ should be $W_s^s (P,Q)$?


**Main Review:**

The paper is original and clearly written. The new connection between soft clustering ensemble and $k$-sparse discrete Wasserstein barycenter is interesting, which allows using techniques in DWB literature to solve the SCE problem. Building a bridge between seemingly different tasks is always a good thing, as it may introduce new tools for both problems. Leveraging the new equivalence, the hardness and consensus analysis are reported for the SCE problem by studying the equivalence DWB problem. Approximation algorithms are also provided, though these results are mostly working in theory due to the efficiency issue (see complexity in Theorem 3). So new efficient algorithms for $k$-sparse DWB might be interesting for further research.

Overall, considering the new connection brings fruitful results for the SCE problem, I tend to vote yes for its acceptance.


**Time Spent Reviewing:**

5

---

> ### Author Response · Authors · 2021-08-10
> **Response to Reviewer Jegc**
>
> We thank the reviewer for the thoughtful comments and suggestions. Our responses and clarifications are summarized below.
>
> **---“The algorithm in Section 4 might run slowly in practice due to the number of permutations ……”**
>
> We agree that the algorithm proposed in Section 4.1 runs slowly in practice (that’s why it is called a “fixed-parameter algorithm”). In experiments, we run the alternating minimization WB algorithm together with the sampling idea (line 342-343), which is more practical and it does not need to enumerate the permutations.
>
> **---“$\mathcal{W}_s(P,Q)$ should be $\mathcal{W}^s_s(P,Q)$”**
>
> Yes, we will change it and thanks for reminding us.

---

### Official Review · Reviewer_jwX5 · 2021-07-18

**Rating:** 6
**Confidence:** 3

**Summary:**

This paper proposes a new approach for soft clustering ensemble, using a new problem formulation (a “geometric prototype”), which is equivalent to a Discrete Wasserstein Barycenter problem.

The proposed algorithm achieves, with a fixed probability 1 – δ, a (1+ ε)-approximate solution to the soft clustering ensemble problem, when the number of clusters k is fixed and small. As the authors state, this is a theoretical result since the runtime is more than exponential in k, 1/ δ, and 1/ε.

In addition, the authors propose an improvement on the existing alternating minimization algorithms in the case of k not being a constant. Then, they prove that the obtained solution converges to the ground-truth clustering as the number m of clustering increases.

Finally, they evaluate the performance of the alternating minimization Wasserstein barycenter algorithm with the proposed improvement, and they compare it to three baselines from the literature. The results show that their algorithm outperforms the baselines in terms of the objective function value and, for the largest datasets, also in terms of runtime.


**Limitations And Societal Impact:**

It's ok.

**Main Review:**

The paper is generally clear and well organized. Occasionally, the writing should be improved, and some small mistakes should be corrected.

The problem is well described and motivated. However, some claims lack adequate reference, as for instance in lines 128-129, lines 138-139; and Lemma 1 and 2. Even if the online supplement contains additional details, the main paper should be self-contained.

The main results appear to be an improvement on an existing algorithm, and the proof of the convergence of the solution with m clustering to the ground truth under weaker assumptions and with a quantitative analysis.

The experimental results are good when compared to the baselines.

A remark: the Discrete Barycenter Problem is NP-hard when the support points of the barycenter are not given or they are not in polynomial size. If the support points of the barycenter are given, problem (6) is not NP-hard, but it is polynomially solvable by Linear Programming, and it can be efficiently solved to optimality also in practice by the barrier algorithm of commercial solvers such as Gurobi (at least for dataset of the size you’re considering), as shown, for example, in

-	Auricchio, G., Bassetti, F., Gualandi, S. and Veneroni, M., 2019. Computing Wasserstein Barycenters via Linear Programming. In CPAIOR-2019, 355-363. Springer.

Questions for the authors: are the baselines method selected for the comparison really the state of the art for the (small) datasets you have used in your experiments? Do you have any test where you have the “real” ground truth, and you can really check that with your approach you are able to recover the true optimal clustering? Evaluating the clustering only in terms of the objective function value can be misleading if the objective function is not the best possible to recover the ground truth.

Some comments:
-	Equation (5) is wrong: it misses a “^1/s” for the minimal value. If you use s for the norm, maybe is better to use the index p for the power of the norm, and “1/p” for the minimum distance.
-	Line 63: that whether; ?
-	Line 68: infinity -> infinite
-	Line 108: C subset of [0,1]^n is not correct (actually S_j is a subset of [0,1]^n)
-	Line 153: formal definition of F is not correct: the conditions on f_ij should be inside the braces
-	Line 170: covert -> convert
-	Section 4.1: the letter of the dimension of the space changes from d to n
-	Algorithm 1: δ, ε >; 0 (instead of ≥)
-	Line 262: that
-	Line 264: since ρ is a density, ρ(C) is not the probability of obtaining C
-	Line 279: what
-	Line 284: necessary to be -> necessarily
-	Assumption 2, as it is formulated, is weaker than Assumption 1: it can be easily seen by taking ξ = L. Can it just be removed?
-	Line 326: the


**Time Spent Reviewing:**

10 hours

---

> ### Author Response · Authors · 2021-08-10
> **Response to Reviewer jwX5**
>
> We thank the reviewer for the thoughtful comments and suggestions. Our responses and clarifications are summarized below.
>
> **----“However, some claims lack adequate reference,…..”**
>
> Thanks for mentioning these to us, and we will add more references to our paper.
>
> **----“are the baselines method selected for the comparison really the state of the art ……………”**
>
> We use three baselines from the papers [25,28,60], which are also well known or recently proposed in the area of clustering ensemble (to our best knowledge). Actually, we also tried several other existing methods (e.g., some methods mentioned in Section 1), but they either have high time complexities or have worse clustering performances than the three.
>
> **----“Do you have any test where you have the “real” ground truth, …..”**
>
> Yes, we considered the comparison with the ground truth in our experiments. We compute the Wasserstein distance between our obtained clustering solution and the ground truth, which indicates the their difference; we show the convergence of this Wasserstein distance as the number of base clusterings (i.e., the value “m”) increases in Figure 2(g), (h), (i). We will add more detailed discussions on this part, and thanks for this question.
>
> **----Other comments.**
>
> Thanks for the suggestions, and we will carefully improve our writing following these helpful comments.

---

> > ### Comment · Reviewer_jwX5 · 2021-08-24
> > **Acknowledge the author's response**
> >
> > We thank the authors for their answers.

---

### Official Review · Reviewer_9VVF · 2021-07-22

**Rating:** 6
**Confidence:** 3

**Summary:**

This manuscript shows the connection between soft clustering ensemble and the discrete Wasserstein barycenter problem. Based on the connection, a sampling-based algorithm is proposed with some provable quality guarantees. The manuscript provides some theoretical analysis about the quality of the obtained consensus clustering results.

**Limitations And Societal Impact:**

Does not apply.

**Main Review:**

This manuscript shows theoretical analyses of soft clustering ensemble problems. It has been known that clustering ensemble can be formulated as a geometric prototype problem. One of the main contributions of this manuscript is to show that the geometric prototype is actually equivalent with an instance of discrete Wasserstein barycenter (DWB). The explanations about this point and intuitions are well described in the manuscript.

A sampling-based algorithm is proposed with provable quality guarantees based on the connection between the soft clustering ensemble and the discrete Wasserstein barycenter. It has been shown that the obtained DWB can be close to the ground-truth clustering if the number of given clustering solutions is large enough. I did not check all the details about the proofs, but the high-level intuition seems reasonable.

I wonder how the proposed method compares to the following paper if the assignment is assumed to be a hard clustering.
* Spectral ensemble clustering (KDD 2015) by H. Liu et al.

I might have missed something, but I wonder how the proposed method can achieve better runtime than the other methods in the experiments for $k=10$ (USPS and CIFAR-10 datasets) even though in Algorithm 1, the third step includes computing $k!$ permutations for $\pi_i$.

In the experiments, it is stated that "we apply random projections to generate the clustering solutions (in each random subspace, we use $k$-means to cluster the data)." More details should be provided to understand this experimental setting clearly. How are the base soft clustering results generated?


**Time Spent Reviewing:**

4

---

> ### Author Response · Authors · 2021-08-10
> **Response to Reviewer 9VVF**
>
> We thank the reviewer for the thoughtful comments and suggestions. Our responses and clarifications are summarized below.
>
> **----“I wonder how the proposed method compares to the following paper……”**
>
> Thanks for mentioning this paper, and we will conduct more experiments with it in our paper.
>
> **----“……..but I wonder how the proposed method can achieve better runtime than the other methods in the experiments for  k=10…..”**
>
> The k! permutations are computed only in our first algorithm (i.e., Algorithm 1 in section 4.1). We admit this algorithm is not that practical (that’s why it is called a “fixed-parameter algorithm”), but it inspires our speedup for the existing DWB algorithms with provable quality guarantee (please see lines 72-75). In our experiments, we apply our sampling idea of Section 4.2 to the alternating minimization WB algorithm (line 342-343). We do not need to enumerate all the k! permutations (in each iteration, the temporary barycenter from the last iteration can determine the matchings).
>
> **----“……..How are the base soft clustering results generated? ”**
>
> We just follow the recent soft clustering ensemble paper [60]’s experimental part to generate the base soft clustering results (by adding random noise to the cluster labels). Thanks for asking this question, and we will explain it in our paper.

---

### Decision · Program_Chairs · 2021-09-27

**Decision:**

Accept (Poster)

**Comment:**

The majority reviewers are in favor of accepting this paper.  The reviewers in general liked the connection made between the soft clustering ensemble problem and discrete Wasserstein barycenter.  The hardness result and convergence analysis rounded out the paper.  However, there was concern about the lack of direct practical implications of the result due to the exponential dependence on the number of centers.